# Can Plasma Volume Determination in Cirrhosis Be Replaced by an Algorithm Using Body Weight and Hematocrit?

**DOI:** 10.3390/diagnostics14080835

**Published:** 2024-04-17

**Authors:** Martine Prütz Nørskov, Thormod Mønsted, Nina Kimer, Morten Damgaard, Søren Møller

**Affiliations:** 1Center of Functional and Diagnostic Imaging and Research, Department of Clinical Physiology and Nuclear Medicine 260, Copenhagen University Hospital, 2650 Hvidovre, Denmark; thormodmrasmussen@hotmail.com (T.M.); morten.damgaard.01@regionh.dk (M.D.); soeren.moeller@regionh.dk (S.M.); 2Gastro Unit, Medical Division, Copenhagen University Hospital, 2650 Hvidovre, Denmark; nina.kimer@regionh.dk; 3Department of Clinical Medicine, Faculty of Health and Medical Sciences, Copenhagen University, 2200 Copenhagen, Denmark

**Keywords:** calculated plasma volume, plasma volume, cirrhosis, portal hypertension

## Abstract

Background: Patients with cirrhosis often develop hyperdynamic circulation with increased cardiac output, heart rate, and redistribution of the circulating volume with expanded plasma volume (PV). PV determination is part of the evaluation of patients with cirrhosis, but gold-standard methods are invasive, expensive, and time-consuming. Therefore, other estimations of PV would be preferable, and the aim of this study was therefore to study if PV, as assessed by a simplified algorithm based on hematocrit and weight, can replace the gold-standard method. Methods: We included 328 patients with cirrhosis who had their PV assessed by the indicator dilution technique as the gold-standard method (PV_I-125_). Actual PV was estimated as PV_a_ = (1 − hematocrit)·(a + (b·body weight)). Ideal PV was estimated as PV_i_ = c · body weight, where a, b, and c are constants. Results: PV_I-125_, PV_a_, and PV_i_ were 3.99 ± 1.01, 3.09 ± 0.54, and 3.01 ± 0.65 (Mean ± SD), respectively. Although PV_I-125_ correlated significantly with PV_a_ (r = 0.72, *p* < 0.001), a Bland–Altman plot revealed wide limits of confidence. Conclusions: The use of simplified algorithms does not sufficiently estimate PV and cannot replace the indicator dilution technique.

## 1. Introduction

The prevalence of cirrhosis is increasing worldwide because of various etiologies [1]. These include, among others, alcohol intake, viral hepatitis, and metabolic-associated steatotic liver disease (MASLD), which denotes a condition with excess fat in the liver [2,3,4]. The prevalence of MASLD is increasing worldwide [5]. In 25% of the patients who are diagnosed with MASLD, this may progress to its more severe form: metabolic-associated steatohepatitis (MASH), and >25% of these patients progress to cirrhosis, following the activation of inflammatory and fibrotic processes [6]. MASLD is associated with obesity, type 2 diabetes, and metabolic syndrome and represents a considerable and increasing health burden [3,7,8]. In the near future, MASLD cirrhosis is expected to become the most common cause of liver transplantation [9]. Patients who develop cirrhosis are at risk of developing portal hypertensive complications such as esophageal varices, hepatic decompensation with the development of fluid retention and ascites, renal failure, and cardiovascular dysfunction. Many of these complications are life-threatening and require intensive care assistance [10]. The Child–Turcotte–Pugh classification system can be used to categorize patients with cirrhosis based on the severity of their liver failure. The Child–Turcotte–Pugh classification system is a universal scoring system that consists of three categories: Child–Pugh class A: good hepatic function, Child–Pugh class B: moderately impaired hepatic function, and Child–Pugh class C: advanced hepatic dysfunction. The scoring system uses both clinical and laboratory data to score the patients [11,12]. 

Patients who are diagnosed with both cirrhosis and portal hypertension possess a poor prognosis, which is largely determined by the degree of portal hypertension, the frequency and severity of complications regarding the liver, and hemodynamic derangement [13,14,15]. 

From a pathophysiological point of view, bacterial translocation, systemic inflammation, and immune dysfunction, despite the active reticuloendothelial system, may trigger fibrogenesis within the liver [16,17,18,19]. This may help contribute to accentuating the splanchnic and systemic hemodynamics derangement leading to multi-organ failure [20]. The main result is peripheral and splanchnic arterial vasodilatation, which can lead to hyperdynamic circulation [21,22], an increased cardiac output, heart rate, blood volume, and plasma volume (PV). A reduction in systemic vascular resistance, arterial blood pressure, and central effective blood volume [1,3,23,24] characterizes this cardiovascular dysfunction. 

The intravascular compartment is known as PV, and the extravascular compartment is known as the fluid within the interstitial space [25]. Patients with fluid retention and ascites, as well as expanded and dysfunctional distributed PV, are important elements in sodium–water retention and circulatory disturbances [26]. Consequently, a clinically applicable technique is important to assess absolute PV levels and for repeated determinations to evaluate disease progression and therapy. 

In daily practice, the gold standard for measuring PV is by using the indicator dilution technique with an injection of a dosage of a radioactive tracer, for example, 125-I-albumin (PV_I-125_) [27]. Although the body equivalent dose is small, less than 0.04 mSv, the procedure is expensive and demanding for both patient and staff since it depends on the placement of a venflon and the regular extraction of blood samples from the contralateral arm. 

PV is related to several variables, including body weight, hematocrit, ejection fraction, and salt intake. A recent study [28] indicated that in healthy individuals and patients with chronic heart failure, PV can reliably be estimated by using the Kaplan–Hakim formula derived by a curve fitting, where actual PV (PV_a_) is calculated by using the patient’s weight and hematocrit value [29]; the ideal PV (PV_i_) formula is based only on weight [30]. These findings have afterward been confirmed in several other large-scale populations of patients with heart failure [31,32]. Hence, it is possible that this method may also be used in patients with cirrhosis and thereby represent an easier way to measure PV. From a clinical perspective, the evaluation of PV in patients with cirrhosis is not typically incorporated into the standard care for uncomplicated cirrhosis. Nevertheless, in specific complex cases, such as those with diuretic-resistant ascites, patients being considered for transjugular intrahepatic portosystemic shunt insertion (TIPS), and patients requiring careful timing for liver transplantation, the assessment of PV may offer valuable insights.

The aim of this study was therefore to investigate if an estimation of PV in patients with cirrhosis can be achieved using the Kaplan–Hakim formula compared with PV_I-125_.

## 2. Materials and Methods

### 2.1. Study Population

In this retrospective study, we included 328 patients with diagnosed cirrhosis. The diagnosis was based on liver biopsy or accepted clinical criteria. All patients with cirrhosis were consecutively referred for a liver vein catheterization and a hemodynamic assessment on suspicion of portal hypertension. Forty-nine patients were excluded due to missing data on PV_I-125_, weight, hematocrit, or failure to cope with histological or clinical criteria of cirrhosis Figure 1.

All patients participated after giving their informed and signed consent, in accordance with the Helsinki II Declaration, and the study was approved by the local Ethics Committee for Medical Research in Copenhagen (journal No. H-17005081 and the Danish Data Protection Agency (AHH-2017-050-05601)). 

### 2.2. Liver Vein Catheterization

After an overnight fasting and a one-hour rest period, catheterization of the hepatic veins and the right atrium of the heart was performed under local anesthesia using a Swan–Ganz catheter (size 7F), which was guided to the hepatic veins and right atrium via the femoral vein using fluoroscopy control. A capacitance transducer (Simonsen & Weel, Copenhagen, Denmark) was used to measure the pressures in the wedged and free hepatic vein position in at least three different positions (the midaxillary line representing the zero-pressure level). Hepatic venous pressure gradient (HVPG) was calculated as the wedge minus free hepatic venous pressure. Mean values of repetitive measurements were used. An HVPG above 5 mmHg was defined as elevated [33].

The systolic, diastolic, and mean arterial blood pressures were measured directly using a small indwelling polyethylene catheter placed in the femoral artery by the Seldinger technique and advanced to the aortic bifurcation. All the pressures were expressed in mmHg. Heart rate was determined by electrocardiography. 

Cardiac output was measured by the indicator dilution technique (IFE IT. 205, Institute of Energy Technique, London, UK), followed by arterial sampling [34]. 

### 2.3. Plasma Volume Measurement

PV_I-125_ was assessed using the indicator dilution technique with the injection of I-125-labelled human serum albumin [27]. Briefly, patients were supine for 30 min, whereafter, a bolus of 185–200 kBq of 125-I-human serum albumin was injected. The blood sample for determination of the PV was collected from the contralateral arm after at least 10 min. All patients were offered a thyroid-blocking dose of 400 mg of potassium iodide after the injection. Volumes are reported as absolute values in L. 

### 2.4. Plasma Volume Equations

We calculated the PV_a_ and PV_i_ using the following equations, which have been previously validated [29,30].
PV_a_ = (1 − haematocrit) · (a + (b · BW))
where a equals 1530 in males and 864 in females, b equals 41.0 in males and 47.9 in females, and BW is the body weight in kg. Hematocrit is applied as a fraction. 

We calculated ideal PV_i_ using the following formula where PV is based on body weight in kg.
PV_i_ = c · BW,
where c equals 39 in males and 40 in females. 

### 2.5. Statistics

Data are presented as mean ± SD. Paired *t*-test/Wilcoxon-test was applied to assess differences in PV’s. Relationships between individual variables were assessed using Spearman’s rank correlation test. Data were normally distributed; see Figure 2. The relation between PV_I-125_ and PV_a_ was evaluated using the Bland–Altman plot. The relation between the different Child–Turcotte–Pugh classifications in PV_I-125_ and PV_a_ was also evaluated using Bland–Altman plots. 

All reported *p*-values were two-tailed, with values less than 0.05 considered significant. Data were analyzed using the program R Studio.

## 3. Results

The clinical, biochemical, and hemodynamic characteristics of the study population consisting of 279 patients with cirrhosis are shown in Table 1. 

The patients’ mean age was 58 [range 31–81 years]. 

Patients had cirrhosis due to different etiologies such as alcohol, cryptogenic liver disease, steatosis, MASH, auto-immune hepatitis, hepatitis B and C, with alcoholic etiology being the most common cause in 209 patients.

According to the Child–Turcotte–Pugh classification, 104 patients belonged to Child–Pugh class A, 86 to Child–Pugh class B, and 89 to Child–Pugh class C. Ascites were present in 141 patients.

The mean PV_I-125_ was higher compared to both PV_a_ and PV_i_ (PV_I-125_: 3.99 ± 1.01 L, vs. PV_a_: 3.09 ± 0.54 L; *p* ≤ 0.001 and vs. PV_i_: 3.01 ± 0.65 L; *p* ≤ 0.001) (Table 2). 

In Child–Turcotte–Pugh class A, B, and C, the mean PV_I-125_ was higher when compared to PV_a_ (Child–Pugh class A: 3.66 ± 0.95 L vs. 3.0 ± 0.56 L; *p* ≤ 0,001, Child–Pugh class B: 3.94 ± 0.88 L vs. 3.0 ± 0.54; *p* ≤ 0.001, Child–Pugh class C: 4.41 ± 1.04 L vs. 3.26 ± 0.47 L; *p* ≤ 0.001) (Table 3).

Spearman’s rank correlation showed a significant correlation between PV_a_ and PV_I-125_ (r = 0.72; *p* ≤ 0.001, Figure 3). 

The Bland–Altman plot between PV_a_ and PV_I-125_ showed a mean difference of 897 mL with a wide confidence interval [95% CI: −545, 2340] (Figure 4A). 

Bland–Altman plots between PV_a_ and PV_I-125_ in the different Child–Turcotte–Pugh classifications all showed wide confidence intervals. In Child–Pugh class A, there was a mean difference of 663 mL [95% CI: −627, 1953] (Figure 4B), there was a mean difference of 917 mL [95% CI: −308, 2144] in Child–Pugh class B (Figure 4C), and a mean of difference of 1152 mL [95% CI: −481, 2787] in Child–Pugh class C (Figure 4D). 

A paired *t*-test revealed with a 95% CI [850, 1109] a mean difference of 979 between PV_I-125_ and PV_a_ with a *p*-value < 0.05; in the patients with ascites; in the patients without ascites, the 95% CI was [691, 921] with a mean difference of 806 with a *p*-value < 0.05. 

## 4. Discussion

The main results of the present study are that although the measured PV_I-125_ correlated with the derived PV_a_, this surrogate marker of PV does not sufficiently and accurately estimate PV and cannot replace the gold-standard technique.

Increased PV is a characteristic feature of the hyperdynamic circulation in cirrhosis [35]. A reduced central blood volume activates renal sodium- and fluid-retaining mechanisms, which leads to the expansion of the PV [36]. Together with portal hypertension and low circulating albumin concentrations, the PV increases with the severity of the disease [35].

Originally, PV was determined by Hamilton in 1928 using an indicator dye. Since then, PV determination has traditionally been based on the indicator dilution technique, either with radioactive-labeled plasma protein or intra-vital dyes that bind to plasma proteins [37]. Using a quantitative injection, the indicator distribution volume is equivalent to the PV plus a small amount of perivascular space, especially in the liver. The gold-standard technique today for PV determination uses I-125 albumin as an indicator [34]. Certain features, including long half-life, thyroid iodine uptake, storage restrictions in the laboratory, and price, may put limitations on the especially repeated use of I-125-albumin [37]. Therefore, alternative indicators or methods should be considered. As an alternative, radioactive 99mTc-albumin has been put forward for the determination of PV [38]. However, it has recently been demonstrated that the use of this indicator consistently overestimates PV [37]. Alternatively, plasma and blood volume can also be measured by labeling red blood cells with 51-Cr [39,40]. However, this method is very time- and staff-intensive. It has therefore been tempting to look at alternative, non-expensive, and more applicable methods such as the use of the Kaplan–Hakim formula. 

Calculation of the PV is, on the contrary, non-invasive, fast, cheap, easy, and does not expose the patients to radioactivity, which is why it would be preferable to replace the more time-consuming, expensive, and demanding PV_I-125_ method. The use of calculated PV has already been proven useful in different patient groups, such as in heart failure and acute respiratory distress syndrome, and to predict outcomes after coronary bypass graft surgery [28,41,42].

To our knowledge, this is the first study to evaluate whether calculated estimates of PV can be used to replace the directly measured gold-standard value of PV in a large group of patients with cirrhosis and portal hypertension. In our study, we found that when PV was calculated using the *Kaplan*–*Hakim* formula with the use of only hematocrit and body weight as parameters, PV was underestimated by 22% compared to PV_I-125_ in the whole patient population. It is noteworthy that PV_i_, which is a simple estimation of the expected optimal PV based on body weight only, on average, is almost 1 L lower than the measured PV_I-125_. This reflects a significant extracellular fluid volume expansion, which, in patients with cirrhosis, may be accumulated primarily as ascites due to splanchnic vasodilatation, higher removal of desaturated albumin, and an increased transvascular escape rate and activated reticuloendothelial system. This distribution of the fluid overload may explain the better performance of the *Kaplan*–*Hakim* formula (PV_a_) in groups of patients with heart failure who usually have no or only minor ascites. Accordingly, a study by Ling et al. [28] that included a smaller cohort of patients with heart failure reported a correlation between directly measured PV_I-125_ and calculated PV_a_ (r = 0.51, *p* = 0.006) with a significantly lower mean bias of −281 mL. In support, we found that the mean bias between PV_a_ and PV_I-125_ increased with the severity of liver damage (and hence increasing ascites) as indicated by the Child–Turcotte–Pugh class (PV_a_ was 18%, 23%, and 26% lower than PV_I-125_ in Child–Pugh class A, Child–Pugh class B, and Child–Pugh class C, respectively). However, a recent larger study by Fudim et al. [32] examined a cohort of 110 patients with NYHA Class III–IV heart failure in hemodynamically steady state condition. They reported a moderate correlation (correlation coefficient of r = 0.64) between directly measured PV and the calculated PV_a_ via the *Kaplan*–*Hakim* formula. Furthermore, they, like the current study, reported an underestimation of PV with a mean bias of approximately 1 L with an even wider 95% CI. Hence, it seems that estimated PV by the *Kaplan*–*Hakim* formula (PV_a_) may perform insufficiently in both patients with heart failure and patients with cirrhosis despite differences in the distribution of the volume overload (ascites). 

In patients with wasted decompensated cirrhosis, the PV is often more expanded. The combination of salt and water retention and sarcopenia may further aggravate the prognosis. Further research should focus on the potential of PV determination in the selection of patients with risk for further decompensation. 

If we could include ascites as a variable in the equation, we might have been able to account for additional fluid volume and provide a more comprehensive assessment of PV. When considering ascites as a factor in the equation, it is important to carefully evaluate its contribution to the overall PV and how it interacts with other variables such as hematocrit and body weight. Ascites can vary in volume and composition among individuals, so incorporating them into the equation may require additional considerations and adjustments to ensure accurate calculations. Overall, including ascites as a factor in the equation for PV assessment has the potential to improve the precision of the measurement and provide a more thorough evaluation of an individual’s fluid status. Further research and validation studies may be necessary to determine the optimal way to incorporate ascites into the equation and assess its impact on the accuracy of PV calculations.

Evaluation of the algorithm-based PV_a_ compared to the radioactive tracer technique (PV_I-125_) is naturally also limited by the imprecision of the latter technique. This includes measurement impression (timing, sample volume, number of samples, etc.) and validity of basic assumptions, for example, the clearance rate of the tracer. Regardless of these considerations, this method is considered the gold standard in clinical practice today.

It should be noted that despite the limited agreement between the calculated and measured absolute PV, several studies indicate that simply calculated relative estimates of intravascular congestion (e.g., PV_a_–PV_i_) are strong predictors of clinical outcome in patients with heart failure [28,43]. This may also be the case in patients with liver cirrhosis. 

In conclusion, calculating PV with the presented calculation will underestimate the actual PV when compared to the gold standard. These findings indicate that the hemoglobin/hematocrit-based formula of PV should be used cautiously if used as an accurate measure of PV. Calculated estimates of PV cannot directly replace the gold-standard assessment of PV.

## Figures and Tables

**Figure 1 diagnostics-14-00835-f001:**
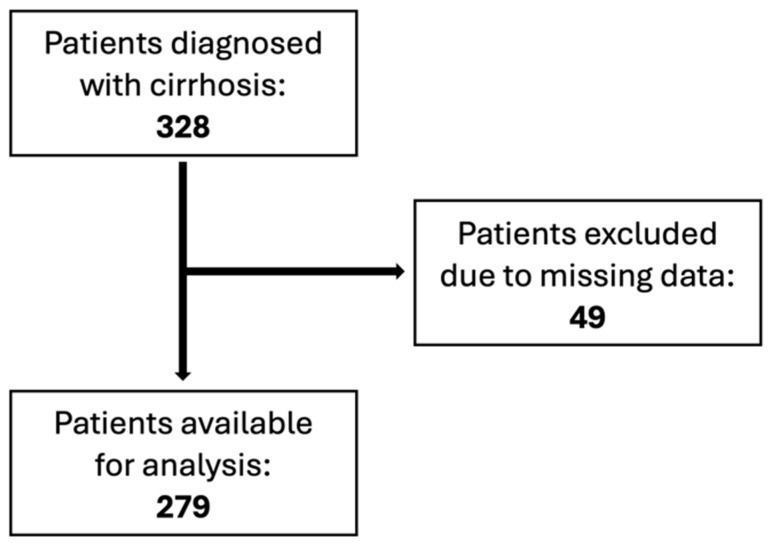
Flowchart illustrating the included patients used for analysis.

**Figure 2 diagnostics-14-00835-f002:**
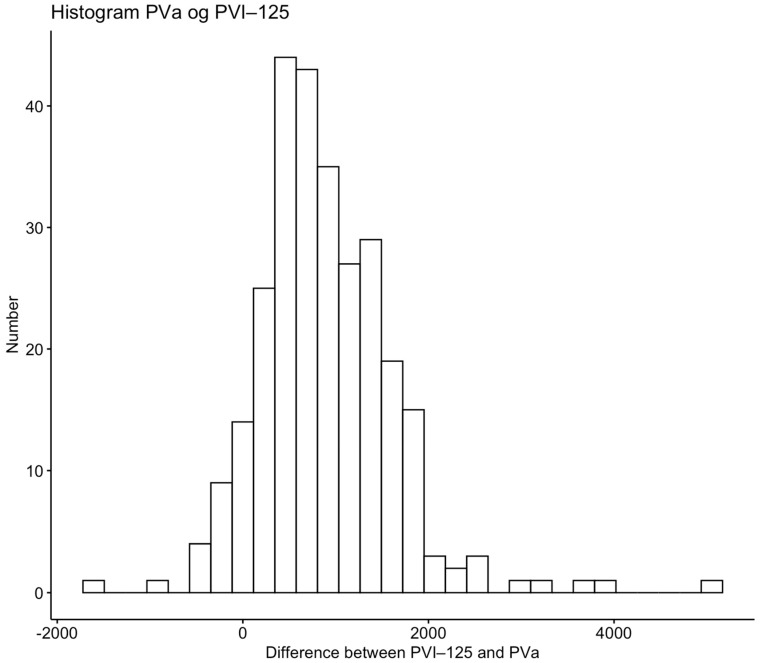
Histogram illustrating a normal distribution of data.

**Figure 3 diagnostics-14-00835-f003:**
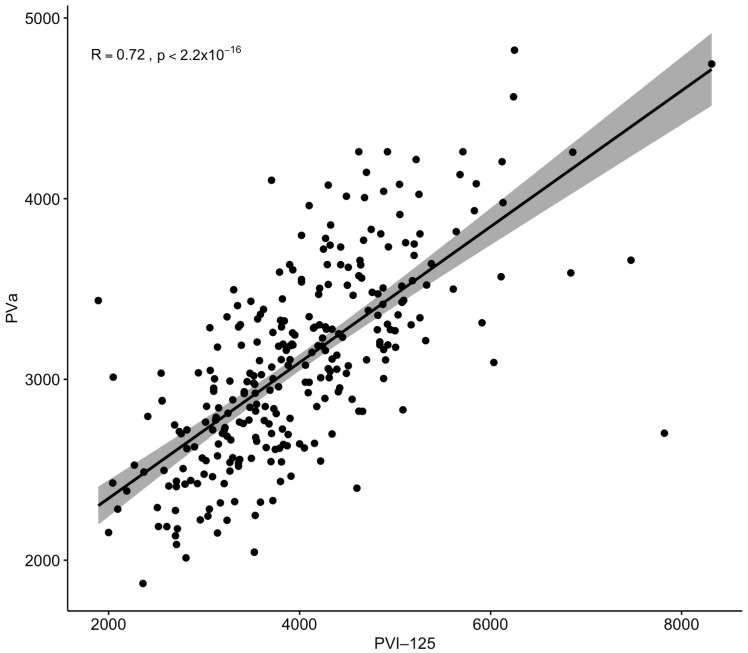
Spearman’s rank correlation between measured PV_I-125_ and calculated PV_a_.

**Figure 4 diagnostics-14-00835-f004:**
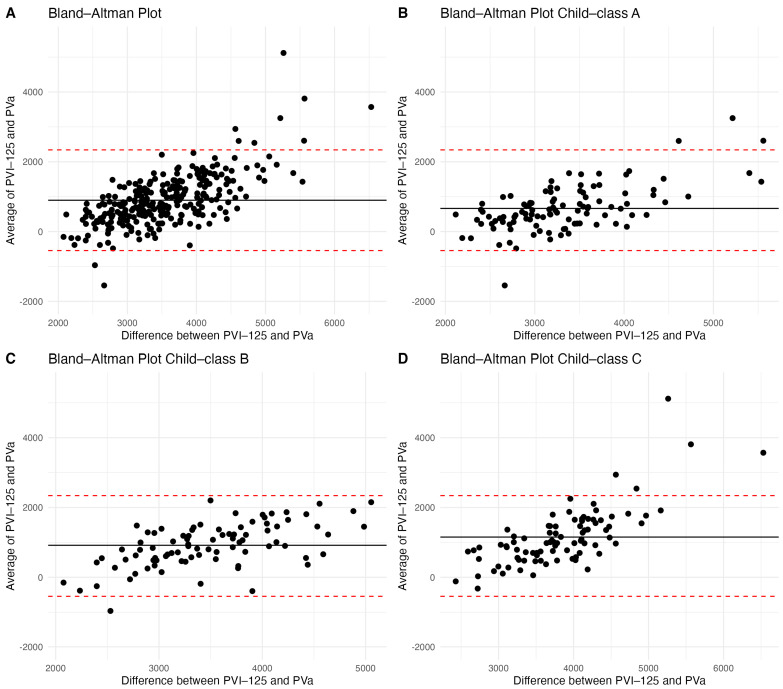
(**A**): Bland–Altman plot for PV_I-125_ and PV_a_. (**B**–**D**): Bland–Altman plots for PV_I-125_ and PV_a_ in the different Child–Turcotte–Pugh Classes: Child–Pugh class A (**B**), Child–Pugh class B (**C**), and Child–Pugh Class C (**D**).

**Table 1 diagnostics-14-00835-t001:** Clinical, biochemical, and hemodynamic characteristics of 279 patients with cirrhosis. Data are shown as mean and standard deviation (SD) [missing data] unless otherwise stated.

Patients Characteristics	
Age (years)	58 (9.4)
Height (cm)	173 (8.5)
Body weight (kg)	77 (16.5)
Gender (M/F)	206/73
Body mass index (Kg/m^2^)	25.2 (4.8)
Ascites (+/−)	141/137 [1]
Esophageal varice (+/−)	75/173 [31]
Coma (+/−)	3/275 [1]
**Biochemistry**	
Blood hemoglobin (mmol/L)	7.5 (1.2)
Serum creatinine (μmol/L)	77.5 (27.9)
Serum sodium (mmol/L)	136.6 (4.6)
Serum Albumin (g/L)	32 (6.1) [3]
Serum Bilirubin (μmol/L9)	23.8 (23.2) [2]
Plasma coagulation factors 2, 7, 10 (U)	0.62 (0.21) [2]
Platelets	169 (96) [3]
ALAT	38.7 (33.5) [3]
INR	1.3 (0.27) [1]
**Hemodynamics**	
Hepatic venous pressure gradient (mmHg)	14.4 (5.9) [3]
Mean arterial pressure (mmHg)	91.5 (14.5)
Heart rate (bpm)	74.6 (14.2)
**The curse of liver cirrhosis stated as the number and (%)**	
Alcohol	209 (74.9)
Alcohol + Hepatitis	3 (1.1)
Alpha 1-antitrypsin deficiency	3 (1.1)
Autoimmune hepatitis	4 (1.4)
Hepatitis B	7 (2.5)
Hepatitis C	16 (5.7)
Hepatitis B, C, and HIV	1 (0.4)
Granulomatous hepatitis	1 (0.4)
Hemochromatosis	1 (0.4)
Cryptogenic liver disease	12 (4.3)
NASH	6 (2.2)
Primary biliary cholangitis	2 (0.7)
Portal hypertension	3 (1.1)
Prehepatic portal hypertension	1 (0.4)
Missing data	10 (3.6)

**Table 2 diagnostics-14-00835-t002:** PV_I-125_, PV, and PV_i_ in 279 patients with cirrhosis. *p*-values are a result of a paired *t*-test.

4	Mean (SD)	*p*-Value
**PV_a_**	3.09 (0.54)	<0.001
**PV_i_**	3.01 (0.65)	<0.001
**PV_I-125_**	3.99 (1.01)	

**Table 3 diagnostics-14-00835-t003:** PV_I-125_ and PV_a_ categorized in the Child–Turcotte–Pugh classification system. *p*-values are a result of a paired *t*-test.

Variable	Mean (SD)	*p*-Value
**PV_a_ Child–Pugh class A**	3.0 (0.56)	<0.001
**PV_I-125_ Child–Pugh class A**	3.66 (0.95)	
**PV_a_ Child–Pugh class B**	3.0 (0.54)	<0.001
**PV_I-125_ Child–Pugh class B**	3.94 (0.88)	
**PV_a_ Child–Pugh class C**	3.26 (0.47)	<0.001
**PV_I-125_ Child–Pugh class C**	4.41 (1.04)	

## Data Availability

The original contributions presented in the study are included in the article, further inquiries can be directed to the corresponding author/s.

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
