# Peer review of "Can Plasma Volume Determination in Cirrhosis Be Replaced by an Algorithm Using Body Weight and Hematocrit?"

_diagnostics, 2024, doi:10.3390/diagnostics14080835_

Round 1
Reviewer 1 Report
Comments and Suggestions for Authors
The Authors performed a comparison analysis between invasive measurement of plasma volume versus surrogate algorhitms in cirrhotic patients. The analysis did not support the use of non-invasive tests in this setting.
The concept is nice and the methods are clearly explained and conducted.
I have some points of concerns:
- please remove the prediction concept from the paper, which cannot be assessed by the analysis performed
- please remove Table 1 of CP classification, which is well acknowledged in literature
- please explain in the introduction the background and results of PVi and PVa that are used in the analysis
- Please do not report any results in the method section, but rather report them in the result section
- Is this study retrospective? a flow chart would be useful to address all inclusion and exclusion criteria of the study
- P-values of the Altman plots should be reported
- Table 2 should report other complication than ascites, etiology of cirrhosis in percentages, INR
- Despite acknolweding the importance of CP classification, it referes to different aspects of ACLD including biochemical parameters. It could be interesting to perform additional analysis evaluating ascitic versus non-ascitic patients to see how the scores perform, regardless of CP classification.
- A focus on cirrhotic hydro retention and sarcopenia could be interesting to support the discrepancies between weight-based scores and invasive tests in cirrhotic plasma volume determination.
Comments on the Quality of English LanguageMinor edits
Reviewer 2 Report
Comments and Suggestions for Authors
The study aimed to investigate if plasma volume (PV) as assessed by a simplified algorithm based on hematocrit and weight can replace the gold standard method. The results showed that the use of simplified algorithms does not sufficiently predict PV and cannot replace the indicator dilution technique. Although the study did not find sufficient mode to replace gold standard method, the idea was novel and worthy. The study was overall well designed and conducted. Some minor points are listed as below.
1. The exclusion criteria should be more detailed and the process of participants’ selection should be illustrated.
2. Would it be possible that the algorithms was sufficient with the combination of more parameters? The authors should discuss the possibility.
3. The test for normal distribution of the data should be provided and the nonparametric test should be applied if the data were not comply with normal distribution.
4. The quality of the figures should be improved.
Comments on the Quality of English LanguageModerate editing of English language required
